# Microseismic Source Location Method and Application Based on NM-PSO Algorithm

Ze Liao [1,*] , Tao Feng [1], Weijian Yu [1,2], Dongge Cui [1] and Genshui Wu [3]

1 School of Resource and Environment and Safety Engineering, Hunan University of Science and Technology, Xiangtan 411201, China
2 Hunan Provincial Key Laboratory of Safe Mining Techniques of Coal Mines, Hunan University of Science and Technology, Xiangtan 411201, China
3 School of Mechanics and Civil Engineering, China University of Mining and Technology, Beijing 100083, China
* Correspondence: leo_hnust@163.com

**Abstract:** Microseismic source location is the core of microseismic monitoring technology in coal mining; it is also the advantage of microseismic monitoring technology compared with other monitoring methods. The source location method directly determines the accuracy and stability of the source location results. Based on the problem of non-benign arrays of microseismic monitoring sensors in the coal mining process, a fast location method of microseismic source in coal mining based on the NM-PSO algorithm is proposed. The core idea of the NM-PSO algorithm is to use the particle swarm optimization (PSO) algorithm for global optimization, reduce the size of the solution space and provide the optimized initial value for the Nelder Mead simplex algorithm (NM), and then use the fast iteration characteristics of the NM algorithm to accelerate the convergence of the model. The NM-PSO algorithm is analyzed by an example and verified by the microseismic source location engineering. The NM-PSO algorithm has a significant improvement in the source location accuracy. The average location errors in all directions are (5.65 m, 5.01 m, and 7.21 m), all Within the acceptable range, and they showed good universality and stability. The proposed NM-PSO algorithm can provide a general fast seismic source localization method for different sensor array deployment methods, which significantly improves the stability and result in the accuracy of the seismic source localization algorithm and has good application value; this method can provide new ideas for research in microseismic localization in coal mining.

**Keywords:** microseismic monitoring; source location; Nelder Mead simplex algorithm; particle swarm optimization algorithm

## 1. Introduction

Microseismic refers to tiny vibrations generated by rock fracture or flow disturbance. Rock fracture and crack extension are accompanied by certain stress changes [1–3]. Therefore, microseismic monitoring can locate rock fractures. Source location is the core advantage of microseismic monitoring technology. Scholars at home and abroad have made rich achievements in researching microseismic source location methods. Different scholars have optimized the location method based on the arrival time difference, which is the most widely used source location method. The location method based on arrival time difference uses different sensors to locate the source of the same microseismic event based on the arrival time difference, wave velocity, and three-dimensional coordinates of the vibration wave. The Geiger algorithm is one of the most classical travel time difference methods; it transforms the nonlinear source solution problem into a linear problem [4]. Romney et al. [5] used the distance residuals to construct the objective function and solved the source position, depth and time respectively. A. Prugger et al. [6] introduced the simplex algorithm to the source location; Later, Zhaozhu et al. [7] applied the simplex algorithm to

the source location in Tibet. Some scholars also use microseismic to conduct positioning research on rock hydraulic seepage [8–10]. Many scholars have also combined different methods to locate the source, using the solution of linear positioning and the least square method as the initial value of the Geiger algorithm, which improves the accuracy and speed of the source solution [11,12]. In order to solve the limitation of the local search method, the global search method is applied to the objective function solution process of the source location. Tang Xingguo et al. [13] used the Powell algorithm for earthquake location, which simplifies the solution process. Pei et al. [14] obtained the plane interval velocity model between wells that realized a very fast simulated annealing (VFSA) inversion. Dai Feng et al. [15] proposed a layered velocity positioning model SV, which was optimized by a genetic algorithm. Many scholars also use a combination of two or more positioning algorithms for microseismic positioning to make up for the shortcomings of a single algorithm and improve positioning accuracy. Lv jinguo et al. [16] combined the robust simulated annealing algorithm with the simplex algorithm to improve positioning accuracy and speed effectively. Guo Yinan et al. [17] proposed a mixed location method of mine microseismic sources based on multi-objective particle swarm optimization simulated annealing (MOPSO-SA). Dong longjun et al. [18–20] proposed a positioning method without pre-velocity measurement in order to reduce the impact of the wave velocity model on the source positioning accuracy. DAVUT et al. [21–24] obtained a series of optimized hybrid algorithms based on the NM algorithm and using its excellent local search capability. In addition, some hybrid algorithms, such as GA-PSO [25], GA-SQP [26], hybridizing PSO [27], and AGSA-PS [28], show high accuracy in terms of arithmetic results. In terms of improved algorithms, IWSA [29], a new modified particle swarm optimization algorithm [30], has also obtained more significant results. h-BOASOS algorithm Used to optimize weight and cost of cantilever retaining wall [31,32], MPBOA was designed borrowing concepts of mutualism and commensalism from the SOS algorithm. Improved numerical results and convergence speed [33], ImWOA algorithm Added diversity to the solution, avoiding the low solution accuracy [34]. The projected algorithm (mWOAPR) is utilized to segment the COVID-19 chest X-ray images [35]. Peng kang et al. [36,37] Adopted different methods to improve the problem of a significant error in the arrival of P-wave and improved the accuracy of the source location. Li Jian et al. [38] established a velocity model without velocity measurement and inversion. Thanks to the powerful nonlinear operation level of deep learning, intelligent and automatic mine microseismic location has been realized [39,40].

When the sensors are arranged in benign arrays, the traditional localization method relying on the coefficient matrix can quickly obtain the optimal hypocenter location. The aforementioned least squares method, Geiger algorithm, and NM simplex algorithm are representative of such localization methods. On the contrary, if the sensor layout is a non-benign array, the accuracy of the positioning method relying on the coefficient matrix cannot be effectively guaranteed, and the method is prone to divergence and non-convergence. When the sensors are arranged as non-benign arrays, the swarm intelligence algorithm (global search algorithm) can also find the best microseismic source position and maintain a high positioning accuracy. The usual methods of this kind of algorithm include a Simulated annealing algorithm, PSO algorithm, genetic algorithm, etc. The disadvantage is that the convergence speed is lower than the traditional positioning method.

In actual underground geotechnical engineering applications, due to the complex geological conditions, such as fault fracture zones, joints, fissures, etc. [41,42]. Sensor deployment arrays are not always guaranteed to be benign arrays, and there may be situations where benign arrays cannot be deployed. Therefore, A NM-PSO seismic source localization algorithm combining the NM algorithm and PSO algorithm is proposed based on the Time Difference of Arrival source localization theory. The proposed algorithm uses the global search property of the PSO algorithm to perform a global search for optimization, reduces the size of solution space, and provides optimized initial values for the NM algorithm. Reducing the problem of the NM algorithm is sensitive to the selection of

initial values; it uses the fast iteration property of the simplex algorithm to speed up the convergence of the model and prevent the model from falling into local optimum or divergence problems.

## 2. Time Difference of Arrival (TDOA) Source Location Theory

Microseismic source location is the core of microseismic monitoring, and it is also an important realization goal of microseismic monitoring. Accurate microseismic location results can effectively identify the time and location of microseismic occurrences, accurately determine the fractured coal and rock mass area, and then study the roof fracture mechanism. The positioning principle is shown in Figure 1. Vibration wave produced by underground rock mass fracture [43,44], and the geophone receives the stress wave. According to the received waveform and arrival time difference, the three-dimensional coordinates of the source are calculated by algorithm.

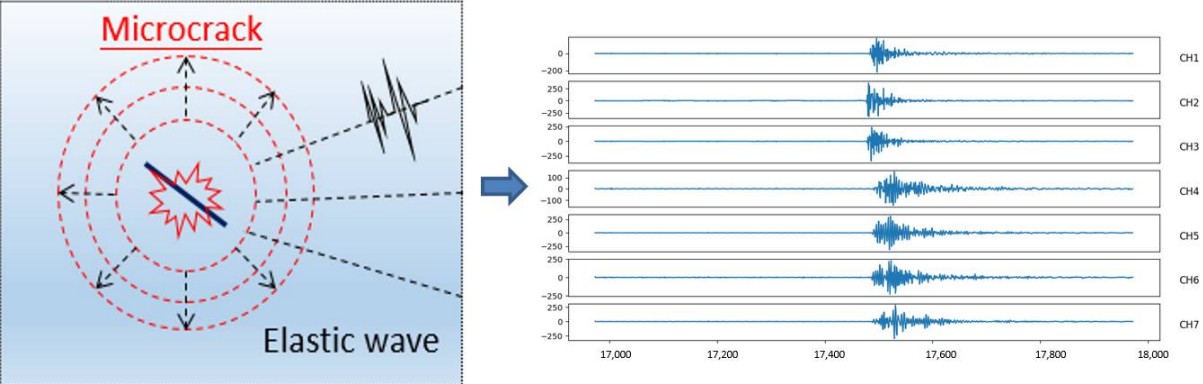

**Figure 1.** Schematic diagram of microseismic positioning principle.

As an inversion problem based on stress wave propagation, microseismic source location can be expressed as:

$$F(m) = D \tag{1}$$

where $D$ is the waveform received by the geophone, the arrival time, the three-dimensional coordinates of the geophone, etc.; $F$ is the stress wave propagation function used, and m is the relevant model parameters, including the location of the source, the propagation medium, and the wave speed. When mining activities are carried out underground, due to frequent stress changes, the amount of signals received by the geophone is large, the frequency of microseismic events is high, the location method based on waveform information is expensive, and the amount of data to be processed is cumbersome, which is not conducive to the quick positioning of underground microseismic events. The source localization method based on arrival time difference is currently the leading research direction in rock engineering. The arrival time difference positioning method can be expressed as:

$$\Delta t = f(r, v, s) \tag{2}$$

In the formula: $\Delta t$ is the time difference data between sensors obtained according to the observed waveform; $f$ is the forward modeling function of stress wave propagation, and the required parameters are the sensor position $r$, the model velocity information $v$, and the hypocenter position $s$.

Based on the above formula, the main factors affecting the location of the source can be summarized as follows:

(1) The accuracy of the time difference information $\Delta t$ between the initial arrival of the stress wave in the observation data;

(2) The accuracy of the model parameters such as the velocity model information v and the performance of the related inversion strategy;

(3)　The influence of the sensor location on the completeness of observational information and the multiplicity of inversion results.

As shown in Figure 2, the source $S(x_0, y_0, z_0, t_0)$ is the unknown parameter that needs to be solved, including the three-dimensional coordinates of the source $S(x_0, y_0, z_0)$ and the seismic time $t_0$, $T_i(x_i, y_i, z_i, t_i)$ is the $i$-th sensor in the microseismic monitoring network, Its three-dimensional coordinates $T_i(x_i, y_i, z_i)$ have been measured in advance, and the first arrival time $t_i$ of the microseismic wave can be obtained by the first arrival picking method.

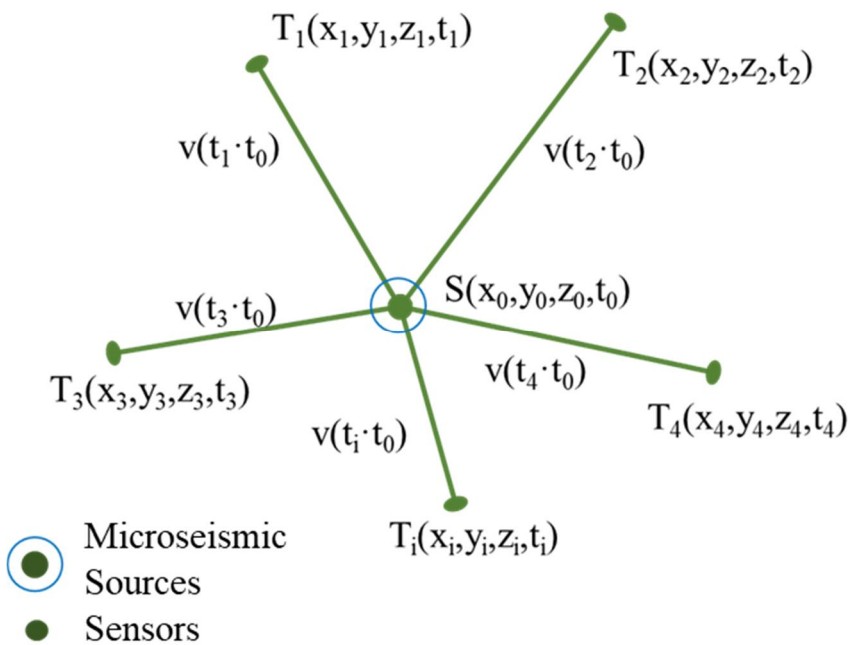

**Figure 2.** Principle of TDOA source location.

Then the calculated travel time $t_i^c$ from the source $S$ to the $i$-th sensor is:

$$t_i^c = \frac{D_i}{v_i(x_i, y_i, z_i)} \tag{3}$$

where $v_i(x_i, y_i, z_i)$ is the propagation velocity of the microseismic wave from the source $S$ to the $i$-th sensor, and $D_i$ is the distance from the source $S$ to the $i$-th sensor.

$$D_i = \sqrt{(x_i - x_0)^2 + (y_i - y_0)^2 + (z_i - z_0)^2} \tag{4}$$

The actual travel time $t_a^i$ from the source $S$ to the $i$-th sensor as follows:

$$t_i^a = t_i - t_0 \tag{5}$$

Then the residua $\gamma_i$ between the actual travel time from the source $S$ to the $i$-th sensor and the calculated travel time is:

$$\gamma_i = t_i^a - t_i^c = t_i - t_0 - t_i^c \tag{6}$$

After the microseismic event occurs, the microseismic wave mainly propagates to the outside world as two spherical waves, P-wave and S-wave. Because P-wave is a longitudinal wave and S-wave is a shear wave, the wave velocity of the P-wave is greater than that of the S-wave. In the research of microseismic underground, S-wave will arrive at the sensor later than P-wave; however, due to the small monitoring range, S-wave is often superimposed at the tail of the P-wave, and it is not easy to separate P-wave and S-wave. Because P-wave arrives first, it is relatively easy to pick up at the first arrival. Therefore,

when a microseismic location is performed underground, the first arrival of the P-wave is used to participate in the source location calculation in most cases. Considering the applicability of microseismic location and the simplified calculation, when microseismic location is performed underground, it is generally assumed that the coal and rock mass is a homogeneous and isotropic medium, and the wave velocity of the vibration wave adopts a fixed value during the propagation process. The residual between the actual travel time and the calculated travel time is as follows:

$$\gamma_i = t_i - t_0 - \frac{\sqrt{(x_i - x_0)^2 + (y_i - y_0)^2 + (z_i - z_0)^2}}{v_p} \tag{7}$$

where $x_0$, $y_0$, $z_0$, and $t_0$ represent the microseismic source parameters, which are unknown parameters to be determined; $x_i$, $y_i$, $z_i$, and $t_i$ represent the three-dimensional coordinates of the *i*-th sensor involved in the calculation and the first arrival of the P-wave, respectively, belonging to known parameters; $i$ = 1, 2, 3, . . . , $n$, where $n$ is the number of effective sensors. When $\gamma_i = 0$, the above equation transforms into:

$$t_i - t_0 = \frac{\sqrt{(x_i - x_0)^2 + (y_i - y_0)^2 + (z_i - z_0)^2}}{v_p} \tag{8}$$

Since there are four unknown parameters to be solved, in order to have a solution to the equation, the number of effective sensors must be $n \geq 4$. When $n = 4$, the equation can theoretically solve each unknown parameter exactly. When $n$ is greater than 4, the equation solution is not unique, and it is generally necessary to use an optimization method to solve the source parameters.

Calculate the error sum of squares of the adequate sensor residuals $\gamma_i$ to get

$$\sum_{i=1}^{n} {\gamma_i}^2 = \sum_{i=1}^{n} \left( t_i - t_0 - \frac{\sqrt{(x_i - x_0)^2 + (y_i - y_0)^2 + (z_i - z_0)^2}}{v_p} \right)^2 \tag{9}$$

By iteratively solving the minimum value of this objective function in the range of the definition domain, the hypocenter parameters can be obtained when $n$ is greater than 4.

## 3. Particle Swarm Optimization (PSO) Algorithm

The PSO algorithm is a swarm intelligence algorithm in intelligent computing; it finds the optimal solution together through sharing population information and the redistribution under the integration of resources.

Assume that the search space is *n*-dimensional. A population $x = (x_1, x_2, \cdots, x_m)$ consisting of m particles, where the position of the *i*-th particle is:

$$x_i = (x_{i1}, x_{i2}, \cdots, x_{in}) \tag{10}$$

Its speed is:

$$v_i = (v_{i1}, v_{i2}, \cdots, v_{in}) \tag{11}$$

The optimal position of the *i*-th particle is:

$$p_i = (p_{i1}, p_{i2}, \cdots, p_{in}). \tag{12}$$

The optimal global solution is:

$$p_g = (p_{g1}, p_{g2}, \cdots, p_{gn}). \tag{13}$$

The quality of the particle position is measured according to the size of the fitness value. After that, the position and speed of the particle will change based on the optimal

position of the individual and the optimal position of the group. Compared with the corresponding optimal position of the group, each iteration will generate a new optimal position to replace the original optimal position, and each particle moves according to the optimal group position obtained by iteration to change its position. At the same time, the movement speed will also change accordingly. After many iterations, the optimal solution to the problem is gradually found. Particles are updated and changed according to the following formula:

$$v_{id}^{k+1} = v_{id}^k + c_1 \cdot rand() \cdot \left( p_{id}^k - x_{id}^k \right) + c_2 \cdot rand() \cdot \left( p_{gd}^k - x_{id}^k \right) \tag{14}$$

$$x_{id}^{k+1} = x_{id}^k + v_{id}^{k+1} \tag{15}$$

where $c_1$ and $c_2$ are called learning factors or acceleration factors; $rand()$ is a random number between (0, 1); $v_{id}^k$ and $x_{id}^k$, respectively, represent particle *i* velocity and position in the *k*-th iteratioof *n* the *d*-th dimension; $p_{id}^k$ is the position of the individual extremum of particle *i* in the d-th dimension; $p_{gd}^k$ is the position of the global extremum of the swarm in the *d*-th dimension.

The speed update equation of the PSO algorithm can be regarded as the following three aspects, among which, $v_{id}^k$ is called the previous speed of the particle; the second part is the cognitive part, the acceleration factor $c_1$ can change the step size, and the main function in this part is to make The particle flies to the individual optimal solution; the third part is usually called the social part, the acceleration factor $c_2$ in this part is different from the cognitive part in the step size adjustment, it is mainly for the adjustment of the optimal global solution.

In order to improve the convergence ability of the PSO algorithm, the optimization of the solution space is more substantial, the inertia weight $\omega$ is introduced into the PSO algorithm, and the above formula is modified as:

$$v_{id}^{k+1} = \omega \cdot v_{id}^k + c_1 \cdot rand() \cdot \left( p_{id}^k - x_{id}^k \right) + c_2 \cdot rand() \cdot \left( p_{gd}^k - x_{id}^k \right) \tag{16}$$

$$x_{id}^{k+1} = x_{id}^k + v_{id}^{k+1} \tag{17}$$

The process of the PSO algorithm is as follows:

(1) Initialize the particle swarm and set various parameters, including the randomly generated initial position and speed;

(2) Select the appropriate fitness function according to the actual problem, and calculate the fitness value of each particle;

(3) Compare the current fitness value of the particle with the optimal historical value. If it is better than the optimal historical value, replace its current position with the best position of the particle; If it is the optimal value of the group, its current position will replace the best position of the group;

(4) Update the particles according to the formula;

(5) If a good enough fitness value is not obtained or the maximum number of iterations is preset, go back to step (2).

## 4. NM Simplex Algorithms

The NM simplex algorithm, proposed by Nelder and Mead in 1965, is designed to perform unconstrained optimization without using gradient information. The operation of this method rescales the simplex according to the local behavior of the function.

To apply the NM algorithm to solve the source location problem, the following steps should be repeated:

Step 1: Initialize

In this step, N + 1 vertices are randomly generated in the search range or space, and the practical value of the objective function is evaluated. The generated vertices are sorted according to the objective function value as follows:

$$NM\_Population = \begin{bmatrix} X_1 & F_1 \\ X_2 & F_2 \\ \vdots & \vdots \\ X_s & F_s \\ X_h & F_h \end{bmatrix}_{(N+1)*2} \tag{18}$$

where $X_i$ is the *i*-th vertex, $X_s$ is the vertex with the second largest objective function value, and F(s) represents the corresponding objective function. $X_h$ and $X_1$ are the vertices with the highest and lowest function values, respectively, while $F_h$ and $F_1$ represent the corresponding observed function values.

Step 2: Mapping

Find the centroid $X_c$ of a simplex that does not contain $X_h$ by Equation (10). A new vertex $X_*$ is generated by mapping the worst point through Equation (11).

$$X_c = \frac{1}{N}\sum_{\substack{j=1 \\ j \neq h}}^{N+1} X_j \tag{19}$$

$$X_* = (1 + \alpha)X_c - \alpha X_h (\alpha > 0) \tag{20}$$

If $F_1 \leq F_* \leq F_s$, replace $X_h$ with $X_*$ and proceed to step 2.

Step 3: Extend

If $F_* \leq F_1$, expand the simplex with an expansion factor $\gamma$ greater than 1 to generate a new vertex $X_{**}$:

$$X_{**} = (1 - \gamma)X_c + \gamma X_* (\gamma > 1) \tag{21}$$

(a) If $F_{**} < F_1$, replace $X_h$ with $X_{**}$ and go to step 2.

(b) If $F_{**} > F_1$, replace $X_h$ with $X_*$ and proceed to step 2.

Step 4: Shrink

If $F_* \geq F_s$, shrink the simplex using the shrinkage factor $\beta$ ($0 < \beta < 1$). There are two types of shrinking rules:

(a) If $F_* < F_h$, generate a new $F_{***}$ with the following formula:

$$F_{***} = (1 - \beta)X_c + \beta X_* (0 < \beta < 1) \tag{22}$$

(b) If $F_* > F_h$, generate a new $F_{***}$ with the following formula:

$$F_{***} = (1 - \beta)X_c + \beta X_h (0 < \beta < 1) \tag{23}$$

Regardless of whether step 4 (a) or step 4 (b) holds, the following two situations need to be considered:

(c) If $F_{***} < F_h$ is satisfied, $F_{***} < F_*$ , $X_{***}$ replaces $X_h$ and then proceed to step 2.

(d) If $F_{***} > F_h$ or $F_{***} > F_*$ is satisfied, replace the size of the simplex by halving the distance from $X_1$, then proceed to step 2.

Step 5: If the stopping condition is met, exit the method; otherwise, continue to step 2.

As a deterministic unconstrained iterative method, the NM simplex algorithm does not require derivation, the calculation is simple, and the iteration speed is fast, but it also has some shortcomings. The simplex algorithm is sensitive to the initial value. When the initial value is unreasonable or the initial range is too large, the simplex algorithm is prone to the problem that the iteration time is too long or falls into the local optimum. Even the maximum side length is long, and the simplex algorithm is prone to problems—a search degradation phenomenon with an area close to zero.

### 5. Construction of NM-PSO Algorithm

The core idea of the NM-PSO algorithm is to perform global optimization based on the global search characteristics of the PSO algorithm. While reducing the size of the solution space, it provides an optimized initial value for the simplex algorithm; it reduces the sensitivity of the simplex algorithm to the selection of the initial value, using the fast iterative characteristics of the simplex algorithm to speed up the convergence of the model and avoid the model falling into local optimum or divergence problems.

The NM-PSO model takes the PSO algorithm as the basic framework. After the population is initialized, the PSO algorithm is used for optimization so that the solution space is reduced in the global scope, and then the particles are sorted according to their fitness, and the first D + 1 particles are composed of Simplex, and then update the particle position through reflection, expansion, compression, etc., calculate the fitness value of each updated particle, and select the particle with the best fitness to replace the optimal particle. The D + 1 particles after simplex transformation and the remaining particles of the original population form a new population again and continue to the following iterative optimization.

The particles with good fitness in the population can converge to the optimal solution in a relatively short time after iterative calculation of the simplex algorithm many times. The iteration speed of the remaining particles without the simplex algorithm in the solution space is relatively slow. While speeding up the solution speed and accuracy of the method can alleviate the population divergence and trap local extreme values to a certain extent.

The source location method based on the NM-PSO algorithm needs to adopt the following steps:

Step 1: In this step, the input data includes the three-dimensional coordinates of the sensors involved in localization, the sensor arrival time, the reflection coefficient $\alpha$, the expansion coefficient $\gamma$, the contraction coefficient $\beta$, the inertia weight $\omega$, and the initial value of the learning factor c, the number of iterations T and the overall number N of the population.

Step 2: In this step, define the objective function of the hypocenter location.

$$\sum_{i=1}^{n} \gamma_i^2 = \sum_{i=1}^{n} \left( t_i - t_0 - \frac{\sqrt{(x_i - x_0)^2 + (y_i - y_0)^2 + (z_i - z_0)^2}}{v_p} \right)^2 \tag{24}$$

Step 3: Initialize the population. Initialize the position and velocity of the particles, calculate the objective function value for each particle, and sort the objective function value for each particle. Record the position of the optimal particle and its function value.

Step 4: Update the position and velocity of each particle, then calculate the fitness value of each particle according to the objective function, and update the global optimal particle and the local optimal particle.

Step 5: Sort the updated particles according to the size of the fitness value, use the NM simplex algorithm to perform simplex transformation on the first D + 1 particles, update the D + 1 particles, and update the optimal particle's location. The remaining particles in the original population continue to update their positions and velocities and update the local optimal particles and the optimal global particles according to the fitness value of the entire population.

Step 6: Determine whether the method satisfies the termination condition, output the result if it is satisfied, and go to Step 4 if it is not satisfied.

The flowchart of the source location method based on the NM-PSO algorithm is shown in Figure 3.

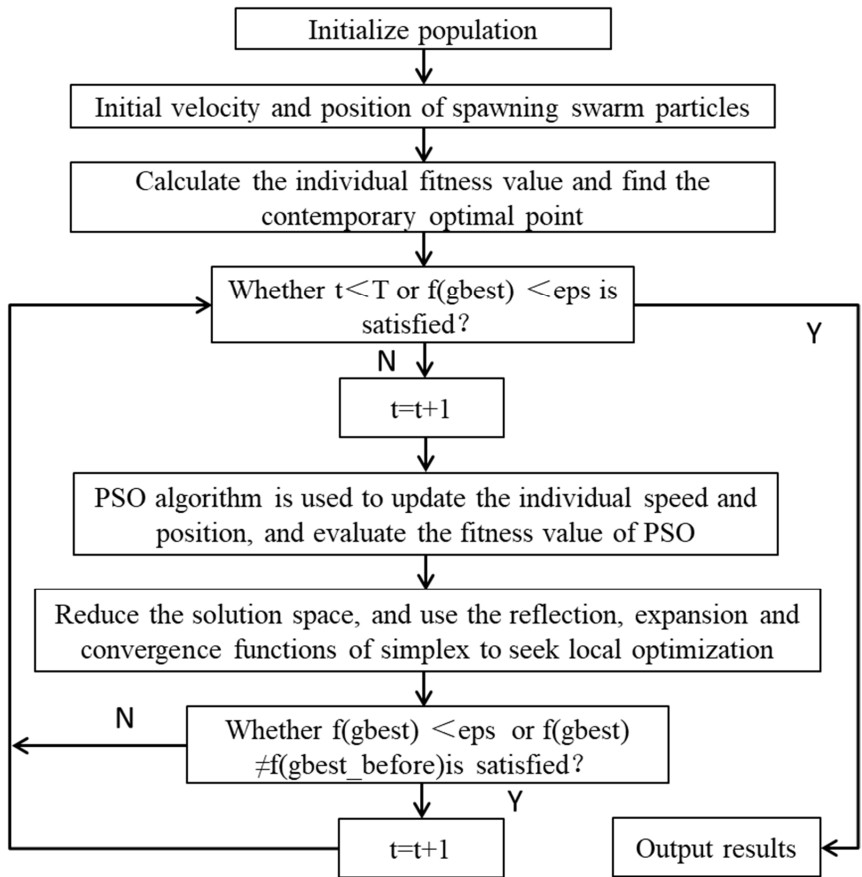

**Figure 3.** Flow chart of NM-PSO algorithm.

## 6. Example Analysis and Verification

In order to verify the feasibility of the proposed NM-PSO source localization algorithm, two example models are set up for benign sensor arrays and non-benign arrays, respectively. As shown in Figure 4, a benign cube array containing eight detectors and a non-benign array of 9 detectors based on the roadway arrangement is set up. Firstly, the seismic source location analysis is carried out with the sensor layout as a benign cube array. As shown in Figure 4a, a total of 8 geophones A, B, C, D, $A_1$, $B_1$, $C_1$, and $D_1$ are set to form a cube array, and the seismic sources are randomly generated. 1, 2, and 3 are the three internal sources of the array, 4 is the source on the array boundary, 5 and 6 are the two external sources of the array, and the coordinates of the sensor and source are shown in Tables 1–3.

**Table 1.** 3D coordinates of benign array detectors.

| Detector Number | X/m | Y/m | Z/m |
|:---:|:---:|:---:|:---:|
| A | 0 | 0 | 0 |
| B | 0 | 1000 | 0 |
| C | 1000 | 1000 | 0 |
| D | 1000 | 0 | 0 |
| $A_1$ | 0 | 0 | 1000 |
| $B_1$ | 0 | 1000 | 1000 |
| $C_1$ | 1000 | 1000 | 1000 |
| $D_1$ | 1000 | 0 | 1000 |

**Table 2.** 3D coordinates of non-benign array detectors.

| Detector Number | X/m | Y/m | Z/m |
|---|---|---|---|
| A | 0 | 0 | 0 |
| I | 100 | 0 | 4 |
| J | 200 | 0 | 0 |
| K | 400 | 0 | 0 |
| L | 500 | 0 | 4 |
| M | 600 | 0 | 0 |
| N | 800 | 0 | 0 |
| P | 900 | 0 | 4 |
| D | 1000 | 0 | 0 |

**Table 3.** 3D coordinates of each source point.

| Source Number | X/m | Y/m | Z/m |
|---|---|---|---|
| 1 | 140 | 460 | 590 |
| 2 | 730 | 380 | 620 |
| 3 | 270 | 840 | 390 |
| 4 | 350 | 1000 | 740 |
| 5 | 540 | 270 | 1200 |
| 6 | 1110 | 640 | 330 |

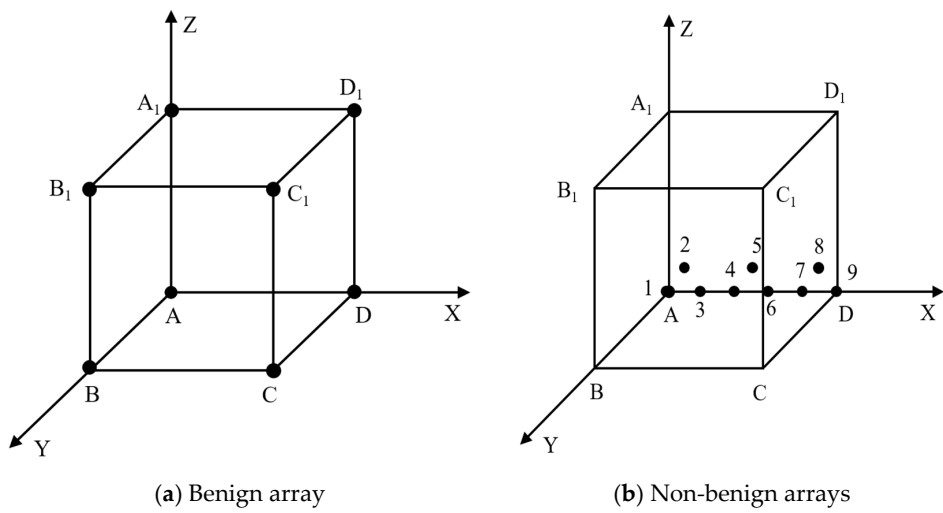

(**a**) Benign array          (**b**) Non-benign arrays

**Figure 4.** Schematic diagram of sensor layout array.

Considering the non-uniformity of stress wave propagation velocity in actual geotechnical engineering, the velocity of each source to each detector is set with a velocity of 5000 m/s plus ±1% disturbance, and the arrival time of each detector are obtained through calculation as shown in Table 4.

In order to verify the effectiveness of the proposed NM-PSO algorithm for the source location problem, two examples of sensor array layout are set up for verification. Compared with the NM simplex algorithm, 24 calculation examples, including two algorithms, six seismic sources, and two sensor arrays have been completed. The positioning analysis of 24 examples is carried out, and the obtained positioning error results are shown in Figure 5.

**Table 4.** Arrival time of each detector (ms).

| Detector | Microseismic Source | | | | | |
|---|---|---|---|---|---|---|
| | i | j | k | l | m | n |
| A | 152.92 | 207.30 | 192.62 | 260.14 | 268.77 | 264.61 |
| B | 163.15 | 230.42 | 99.23 | 163.51 | 300.17 | 241.04 |
| C | 235.63 | 182.73 | 169.57 | 198.32 | 294.94 | 100.57 |
| D | 227.75 | 154.33 | 236.24 | 278.30 | 262.70 | 146.43 |
| $A_1$ | 126.73 | 180.07 | 212.74 | 219.82 | 128.66 | 268.52 |
| $B_1$ | 138.68 | 206.74 | 137.29 | 87.77 | 186.66 | 268.52 |
| $C_1$ | 218.57 | 155.12 | 193.41 | 138.81 | 177.82 | 152.29 |
| $D_1$ | 212.61 | 120.24 | 256.35 | 246.20 | 113.59 | 186.78 |
| A | 153.20 | 205.80 | 192.68 | 260.47 | 268.18 | 265.80 |
| I | 148.65 | 193.61 | 186.92 | 253.95 | 261.40 | 250.11 |
| J | 149.98 | 179.55 | 184.13 | 250.17 | 255.42 | 232.87 |
| K | 157.93 | 160.06 | 187.50 | 249.89 | 247.67 | 201.98 |
| L | 164.77 | 150.75 | 190.18 | 247.75 | 244.62 | 188.36 |
| M | 175.71 | 147.21 | 196.11 | 252.46 | 244.59 | 178.06 |
| N | 199.48 | 145.21 | 213.84 | 264.52 | 250.59 | 155.78 |
| P | 210.93 | 148.03 | 224.50 | 270.72 | 255.32 | 148.58 |
| D | 228.89 | 154.07 | 233.52 | 278.25 | 262.28 | 146.66 |

Note: Because the speed is set with random numbers, there is a certain difference in the arrival times of A and D.

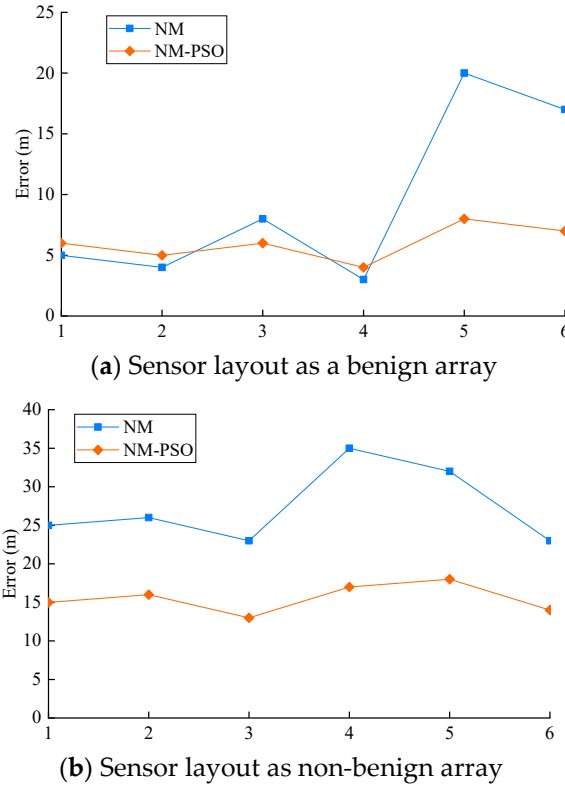

(**a**) Sensor layout as a benign array

(**b**) Sensor layout as non-benign array

**Figure 5.** Comparison of positioning errors.

It can be seen from Figure 5a that when the sensor is arranged as a benign array, both the NM simplex algorithm and the NM-PSO algorithm can obtain a high positioning error, and the maximum error is controlled within 20 m. Further analysis shows that when the hypocenter point is within the envelope of the sensor array, the hypocenter location obtained by the two methods can be used as the result of the hypocenter location; however, when the NM simplex algorithm solves the source coordinates located outside the geophone array, the source positioning error increases significantly. At the same time,

the NM-PSO algorithm can also ensure higher positioning accuracy. The iterative solution of the hypocenter location method based on the NM-PSO algorithm can better locate the hypocenter outside the geophone array.

The reason is that the simplex algorithm is essentially a search-type iterative algorithm, and each iteration will make the result closer to the actual value. The accuracy of the simplex algorithm will gradually improve during the iterative process, and the error will become smaller and smaller; however, the simplex algorithm's accuracy dramatically depends on the initial value selection. Suppose the initial value is not selected correctly. In that case, the method's accuracy will continue to improve in the local area; however, it may fall into local extreme values in the global solution space and affect its final iteration's outcome. The NM-PSO algorithm belongs to the global-local search method; its global search capability can provide more accurate initial values. Thanks to the fast convergence capability of NM simplex, it is ideal when dealing with sources inside and outside the array.

It can be seen from Figure 5b that when an ideal sensor array cannot be arranged due to the actual geotechnical conditions. There will be a significant error in the hypocenter position solved by the simplex algorithm, and the maximum error reaches 35 m. Further analysis shows that the positioning error will increase significantly when the source is outside the sensor array. Whether the NM simplex algorithm or the NM-PSO algorithm, the positioning error is nearly doubled compared to the benign array. Because the source outside the sensor array is too far from the sensor, the arrival time and the error in the iterative process continue accumulating, and the absolute positioning error is amplified geometrically; it also agrees with the actual situation monitored on the spot. When the sensor array is non-benign, the NM-PSO algorithm is insensitive to the initial value setting; it avoids the influence of improper initial value setting on the source positioning error. Compared with the NM simplex algorithm, the positioning accuracy is improved. At the same time, it also has better universality and stability.

## 7. Engineering Verification

In order to further verify the effectiveness of the NM-PSO source location algorithm proposed in this paper, the ZF3808 working face of Shuiliangdong coal mine in Bin County, Shaanxi Province, is selected as the engineering background, and the engineering example is checked and analyzed. Figure 6 shows the mine's microseismic monitoring system.

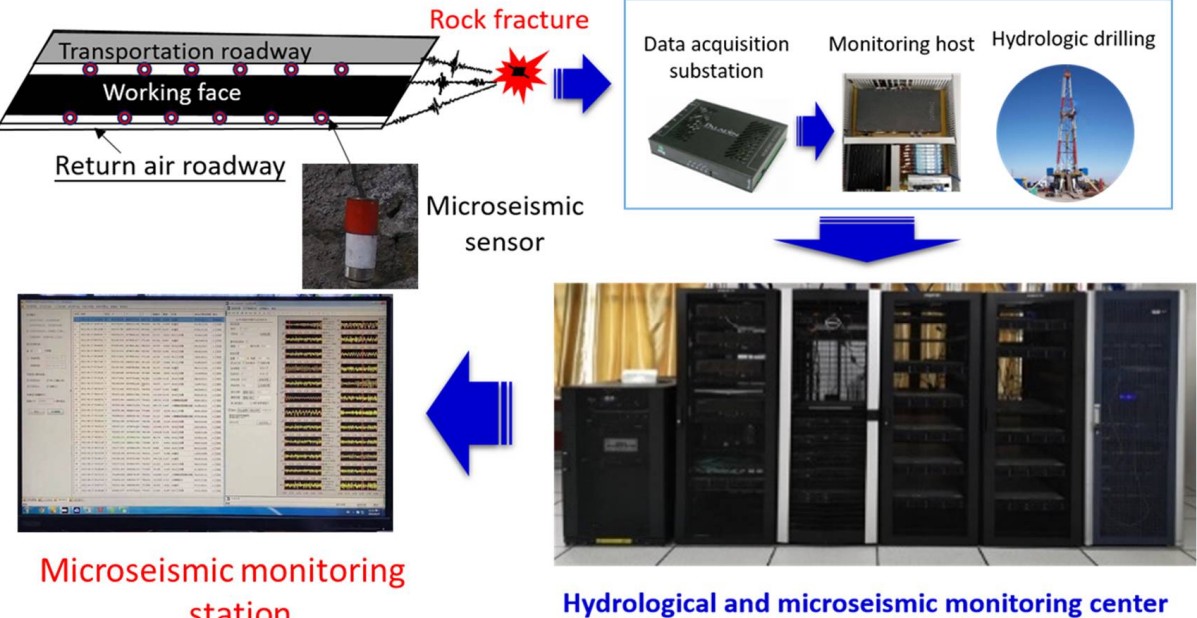

**Figure 6.** Flow chart of microseismic monitoring.

Three known coordinate points are set and knocked to create microseismic events. Then NM simplex algorithm and NM-PSO source location algorithm are used to locate and calculate microseismic events. The positioning results are shown in Table 5; it can be seen from Table 5 that the two positioning methods mentioned in this paper have achieved ideal positioning results in all directions of microseismic events. Compared with the PSO optimization algorithm and NM algorithm, the event location calculated by the NM-PSO source location algorithm proposed in this paper is closer to the known coordinate points, and the overall error control is more stable. The final average location error in each direction is (5.65 m, 5.01 m, and 7.21 m), which is within the acceptable range and can meet the engineering needs. Therefore, the NM-PSO source location algorithm proposed in this paper can be applied to engineering practice.

**Table 5.** Positioning results of each algorithm.

| Number | NM | | | NM-PSO | | | Original Coordinates | | |
|---|---|---|---|---|---|---|---|---|---|
| | X | Y | Z | X | Y | Z | X | Y | Z |
| A | 146.05 | 249.06 | 755.22 | 139.03 | 244.63 | 747.43 | 134.6 | 238.3 | 740 |
| B | 248.69 | 132.21 | 764.54 | 247.97 | 125.81 | 755.66 | 240.7 | 121 | 750 |
| C | 144.49 | 122.68 | 776.78 | 139.16 | 118.79 | 768.54 | 133.9 | 114.9 | 760 |

## 8. Conclusions

Based on the Time Difference of Arrival source localization theory, this paper constructs the NM-PSO algorithm for the location accuracy of microseismic sources in the coal mining process and conducts example analysis and engineering verification. The following conclusions are obtained:

(1) A microseismic source location method based on the NM-PSO algorithm is proposed for different sensor arrays. The algorithm not only retains the technical advantages of the NM algorithm's fast iteration and easy convergence but also avoids the initial value sensitivity problem of the NM algorithm. At the same time, the global optimization characteristics of the PSO algorithm are used so that the NM-PSO algorithm has better universality and stability.

(2) Through the example analysis and engineering verification of the NM-PSO algorithm, it can be concluded that the NM-PSO algorithm avoids the influence of improper initial value setting on the source localization error, and the source location accuracy is significantly improved compared with the NM simplex algorithm. The engineering verification results further prove the accuracy and stability of the method, indicating that the NM-PSO algorithm can meet the engineering needs and achieve the required localization accuracy.

(3) The microseismic source's location in the coal mining process is affected by many factors, among which the anisotropy of the rock body affects the wave propagation speed, and the complex geological formations can cause different sensor arrays, which significantly affect the positioning accuracy. Therefore, in the follow-up research, a more suitable algorithm model can effectively improve the localization accuracy and more accurately realize the microseismic source localization in the coal mining process.

**Author Contributions:** Writing—original draft preparation, Z.L.; Data curation, Z.L. and T.F.; Investigation, Z.L. and W.Y.; Formal analysis, Z.L. and D.C.; Funding acquisition, W.Y.; Methodology, Z.L. and G.W. All authors have read and agreed to the published version of the manuscript.

**Funding:** This research was funded by the National Natural Science Foundation of China (51974117, 51804114) and the Natural Science Foundation of Hunan Province (2020JJ4027).

**Institutional Review Board Statement:** Not applicable.

**Informed Consent Statement:** Not applicable.

**Data Availability Statement:** The data used to support the findings of this study are available from the corresponding author upon request.

**Conflicts of Interest:** The authors declare no conflict of interest.

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
