# Peer review of "Microseismic Source Location Method and Application Based on NM-PSO Algorithm"

_applsci, doi:10.3390/app12178796_

Round 1
Reviewer 1 Report
I have the following comments:
1) The NM and PSO algorithms have already been hybridized in the previously reported works. Therefore, the authors must explain the differences of their proposed algorithm with that of the following to clearly indicate the novelty. Otherwise, it will be just a re-iteration of the previous works.
** Dynamical modeling of Li-ion batteries for electric vehicle applications based on hybrid Particle Swarm–Nelder–Mead (PSO–NM) optimization algorithm
** Hybrid Nelder–Mead simplex search and particle swarm optimization for constrained engineering design problems
** A novel optimal capacitor placement algorithm using Nelder-Mead PSO
2) The other recent examples of the hybridization examples that employ the NM technique must also be discussed in the manuscript, and the difference of the proposed method in this manuscript must clearly be stated.
** Design and application of an optimally tuned PID controller for DC motor speed regulation via a novel hybrid Lévy flight distribution and Nelder–Mead algorithm
** SCANM: A Novel Hybrid Metaheuristic Algorithm and its Comparative Performance Assessment
** A Novel Hybrid ASO-NM Algorithm and Its Application to Automobile Cruise Control System
** An enhanced slime mould algorithm for function optimization
3) The resolution of images must be increased and the presentation language must be improved.
4) The performance of the proposed NM-PSO algorithm must be tested against some of the benchmark functions.
5) In section 6, the analyses performed on two models must also be extended by comparing it with the other available algorithms (apart from NM), as well.
Author Response
Thank you for your comments concerning our manuscript entitled " Microseismic source location method and application based on NM-PSO algorithm." Those comments are valuable and helpful for revising and improving our paper and the important guiding significance to our research in the future. We have studied comments carefully and have made revisions, which we hope to meet with approval. Revised portions are marked in red in the manuscript. The primary revisions in the manuscript and the response to the comments are as following:
Point 1: The NM and PSO algorithms have already been hybridized in the previously reported works. Therefore, the authors must explain the differences of their proposed algorithm with that of the following to clearly indicate the novelty. Otherwise, it will be just a re-iteration of the previous works.
** Dynamical modeling of Li-ion batteries for electric vehicle applications based on hybrid Particle Swarm–Nelder–Mead (PSO–NM) optimization algorithm
** Hybrid Nelder–Mead simplex search and particle swarm optimization for constrained engineering design problems
** A novel optimal capacitor placement algorithm using Nelder-Mead PSO
Response 1: Thank you for your comment. We have carefully studied these three articles, they are very helpful for us. We believe that NM-PSO algorithm has been applied to the study of microseismic source location in coal mining for the first time in our manuscript, and has obtained more accurate results than NM algorithm in our case verification.
Point 2: The other recent examples of the hybridization examples that employ the NM technique must also be discussed in the manuscript, and the difference of the proposed method in this manuscript must clearly be stated.
** Design and application of an optimally tuned PID controller for DC motor speed regulation via a novel hybrid Lévy flight distribution and Nelder–Mead algorithm
** SCANM: A Novel Hybrid Metaheuristic Algorithm and its Comparative Performance Assessment
** A Novel Hybrid ASO-NM Algorithm and Its Application to Automobile Cruise Control System
** An enhanced slime mould algorithm for function optimization
Response 2: We appreciate it very much for this good suggestion, and we have done it according to your ideas. Please refer to the revised manuscript.
Point 3: The resolution of images must be increased and the presentation language must be improved.
Response 3: Thank you for your comment. We have redrawn the pictures in the manuscript to ensure clarity of the pictures. In addition, we uploaded all pictures as an attachment.
Point 4: The performance of the proposed NM-PSO algorithm must be tested against some of the benchmark functions.
Response 4: Thank you for your comment. NM algorithm has been widely used in the study of microseismic source location in coal mining, and its performance has been verified by a lot of engineering practice. At the same time, the PSO algorithm has been tested in reference [1]. Therefore, the NM-PSO algorithm has not been tested against the benchmark functions.
[1]Zhao, W.; Wang, L.; Zhang, Z. Atom search optimization and its application to solve a hydrogeologic parameter estimation problem. Knowl.-Based Syst. 2019, 163, 283-304.
Point 5: In section 6, the analyses performed on two models must also be extended by comparing it with the other available algorithms (apart from NM), as well.
Response 5: Thank you for your valuable and thoughtful comments. However, we believe that in the study of microseismic source location in coal mining, NM algorithm is the most widely used. The primary research objective of the manuscript is to further improve the source location accuracy of NM algorithm through PSO algorithm. Therefore, we have not considered the comparison of other algorithms except NM.
Once again, we thank the time you put in reviewing our manuscript. Your comments have been precious.
Reviewer 2 Report
the manuscript is ok and seems have some novelty.
abstract and conclusion must be more informative
the discussion is very weakly presented.
Author Response
Thank you for your comments concerning our manuscript entitled " Microseismic source location method and application based on NM-PSO algorithm." Those comments are valuable and helpful for revising and improving our paper and the important guiding significance to our research in the future. We have studied comments carefully and have made revisions, which we hope to meet with approval. Revised portions are marked in red in the manuscript. The primary revisions in the manuscript and the response to the comments are as following:
Point 1: the manuscript is ok and seems have some novelty.
abstract and conclusion must be more informative
the discussion is very weakly presented.
Response 1: Thank you for your valuable and thoughtful comments. We have revised the abstract and conclusions and added discussion in the introduction. Please refer to the revised manuscript.
Once again, we thank the time you put in reviewing our manuscript. Your comments have been precious.
Reviewer 3 Report
This study develops the NM-PSO source location method combining the PSO algorithm and Nelder-Mead simplex algorithm. The paper includes new contributions with good merits for publication. However, a number of issues/errors in the manuscript are expected to be solved.
1- The authors may reorganize the abstract to make it reasonable and shed the light on the aim of the proposed work. In addition, the novelty of the paper should be properly addressed in the abstract, the introduction, and the conclusion.
2- The principal gist of this study is not reached easily. The originality and the novelty of the study reflecting the main gist of the paper should more clearly be mentioned in the Introduction section.
3- The review of optimization methods can be expanded in the introduction. Some works are recommended to be added, such as:
Delice, Y., et al. (2017). "A modified particle swarm optimization algorithm to mixed-model two-sided assembly line balancing." Journal of Intelligent Manufacturing 28(1): 23-36.
Khajehzadeh, M., et al. (2014). "Multi-objective optimisation of retaining walls using hybrid adaptive gravitational search algorithm." Civil Engineering and Environmental Systems 31(3): 229-242.
Koessler, E. and A. Almomani (2021). "Hybrid particle swarm optimization and pattern search algorithm." Optimization and Engineering 22(3): 1539-1555.
Eslami, M., et al. (2011). "Damping Controller Design for Power System Oscillations Using Hybrid GA-SQP." International Review of Electrical Engineering-Iree 6(2): 888-896.
Cherki, I., et al. (2019). "A Sequential Hybridization of Genetic Algorithm and Particle Swarm Optimization for the Optimal Reactive Power Flow." Sustainability 11(14): 3862.
Kaveh, A., et al. (2020). "An improved water strider algorithm for optimal design of skeletal structures." Periodica Polytechnica Civil Engineering 64(4): 1284-1305.
4- How the authors define the parameters of the algorithm? Should be discussed.
5- A lot of recent well-known metaheuristic algorithms have been developed. But there is no justification for the newly proposed method.
6- To verify the performance of the algorithm, the author should apply the new method to CEC or GECCO benchmark functions and compare the results with other well-known optimization algorithms.
7- Statistical analysis should be provided for the results and the authors "must" perform hypothesis test and report the p-value over executions of "all" problems.
8- The quality of the figures should be enhanced.
9- The Conclusion section should be expanded in terms of content. Hence, this section should be revised to present concise conclusions as bullet points and/or numbered entries. Namely, the conclusions and fundamental motivations of the study should be added in a clearer way in this section.
Author Response
Thank you for your comments concerning our manuscript entitled " Microseismic source location method and application based on NM-PSO algorithm." Those comments are valuable and helpful for revising and improving our paper and the important guiding significance to our research in the future. We have studied comments carefully and have made revisions, which we hope to meet with approval. Revised portions are marked in red in the manuscript. The primary revisions in the manuscript and the response to the comments are as following:
Point 1: The authors may reorganize the abstract to make it reasonable and shed the light on the aim of the proposed work. In addition, the novelty of the paper should be properly addressed in the abstract, the introduction, and the conclusion.
Response 1: We appreciate it very much for this good suggestion, and we have done it according to your ideas. Please refer to the revised manuscript.
Point 2: The principal gist of this study is not reached easily. The originality and the novelty of the study reflecting the main gist of the paper should more clearly be mentioned in the Introduction section.
Response 2: Thank you for underlining this deficiency. This section was modified according to your comments. Please refer to the revised manuscript.
Point 3: The review of optimization methods can be expanded in the introduction. Some works are recommended to be added, such as:
Delice, Y., et al. (2017). "A modified particle swarm optimization algorithm to mixed-model two-sided assembly line balancing." Journal of Intelligent Manufacturing 28(1): 23-36.
Khajehzadeh, M., et al. (2014). "Multi-objective optimisation of retaining walls using hybrid adaptive gravitational search algorithm." Civil Engineering and Environmental Systems 31(3): 229-242.
Koessler, E. and A. Almomani (2021). "Hybrid particle swarm optimization and pattern search algorithm." Optimization and Engineering 22(3): 1539-1555.
Eslami, M., et al. (2011). "Damping Controller Design for Power System Oscillations Using Hybrid GA-SQP." International Review of Electrical Engineering-Iree 6(2): 888-896.
Cherki, I., et al. (2019). "A Sequential Hybridization of Genetic Algorithm and Particle Swarm Optimization for the Optimal Reactive Power Flow." Sustainability 11(14): 3862.
Kaveh, A., et al. (2020). "An improved water strider algorithm for optimal design of skeletal structures." Periodica Polytechnica Civil Engineering 64(4): 1284-1305.
Response 3: We appreciate it very much for this good suggestion, and we have done it according to your ideas. Please refer to the revised manuscript.
Point 4: How the authors define the parameters of the algorithm? Should be discussed.
Response 4: The main parameters of the algorithm are as follows: Reflection coefficient α, Expansion coefficient γ, Shrinkage coefficient β, Inertia weight ω and The initial value of the learning factor C, The number of iterations T, and The total population number N. The specific parameter assignment is determined according to different research conditions.
Point 5: A lot of recent well-known metaheuristic algorithms have been developed. But there is no justification for the newly proposed method.
Response 5: The purpose of the manuscript is to improve the positioning accuracy of the NM algorithm in the study of the location of microseismic sources in coal mining. The NM algorithm is sensitive to the initial value, and is prone to the problem of too long iteration time or falling into local optimization. The PSO algorithm can reduce the sensitivity of the NM algorithm to the selection of the initial value. Therefore, we choose the PSO algorithm to solve this problem.
Point 6: To verify the performance of the algorithm, the author should apply the new method to CEC or GECCO benchmark functions and compare the results with other well-known optimization algorithms.
Response 6: Thank you for your comment. NM algorithm has been widely used in the study of microseismic source location in coal mining, and its performance has been verified by a lot of engineering practice. At the same time, the PSO algorithm has been tested in reference [1]. Therefore, the NM-PSO algorithm has not been tested against the benchmark functions. At the same time, we believe that the NM algorithm is the most widely used in the study of microseismic source location in coal mining. Our primary research objective is to further improve the source location accuracy of the NM algorithm. Therefore, we have not considered other algorithms other than the NM algorithm for comparison.
[1]Zhao, W.; Wang, L.; Zhang, Z. Atom search optimization and its application to solve a hydrogeologic parameter estimation problem. Knowl.-Based Syst. 2019, 163, 283-304.
Point 7: Statistical analysis should be provided for the results and the authors "must" perform hypothesis test and report the p-value over executions of "all" problems.
Response 7: Thank you for your valuable and thoughtful comments. In this manuscript, the data of the verification results in Section 6 and Section 7 of this manuscript are relatively small, so we did not perform hypothesis testing. However, the hypothesis test you mentioned is very helpful for our subsequent research work. In the follow-up engineering application, there is a large amount of microseismic source data, and we will take the hypothesis test as an important content of the follow-up research. Thanks again for this comment.
Point 8: The quality of the figures should be enhanced.
Response 8: Thank you for your comment. We have redrawn the pictures in the manuscript to ensure clarity of the pictures. In addition, we uploaded all pictures as an attachment.
Point 9: The Conclusion section should be expanded in terms of content. Hence, this section should be revised to present concise conclusions as bullet points and/or numbered entries. Namely, the conclusions and fundamental motivations of the study should be added in a clearer way in this section.
Response 9: Thank you for your valuable and thoughtful comments. We have revised the Conclusion section. Please refer to the revised manuscript.
Once again, we thank the time you put in reviewing our manuscript. Your comments have been precious.
Author Response
Thank you for your comments concerning our manuscript entitled " Microseismic source location method and application based on NM-PSO algorithm." Those comments are valuable and helpful for revising and improving our paper and the important guiding significance to our research in the future. We have studied comments carefully and have made revisions, which we hope to meet with approval. Revised portions are marked in red in the manuscript. The primary revisions in the manuscript and the response to the comments are as following:
Point 1: There are many formatting problems throughout the manuscript. The authors need to fix those.
Response 1: Thank you for underlining this deficiency. We have checked the manuscript carefully and reviewed them. Please refer to the revised manuscript.
Point 2: The motivation is not clear. Why the authors used a metaheuristic algorithm (PSO) to solve this problem. I think the problem can be solved by an analytic method. Without a strong reason for using metaheuristic for this problem, this cannot be considered as a notable work.
Response 2: Thank you for your comment. The purpose of the manuscript is to improve the positioning accuracy of microseismic sources based on the NM algorithm. The NM algorithm is sensitive to the initial value, and it is prone to the problem of too long iteration time or falling into the local optimum, while the PSO algorithm can reduce the problem that the NM algorithm is sensitive to the selection of the initial value. Therefore we choose the PSO algorithm to solve this problem.
Point 3: Add a paragraph to show that metaheuristic algorithms are often used to solve real life problems. You may use the following works for example
https://doi.org/10.1016/j.asoc.2016.09.037
https://doi.org/10.1016/j.compbiomed.2021.104984
https://doi.org/10.1007/s00366-021-01294-x
https://doi.org/10.1007/s11042-020-10053-x
https://doi.org/10.1007/s10462-021-10114-z
Response 3: We appreciate it very much for this good suggestion, and we have done it according to your ideas. Please refer to the revised manuscript.
Point 4: Please use the abbreviation after first use and make them uniform. For example, in abstract section (Line number 20), “for the Nelder Mead simplex algorithm (NM)…” the authors used abbreviation ‘NM’ for ‘Nelder Mead simplex algorithm’. Also, the authors used NM in Line18.
Response 4: Thank you for underlining this deficiency. We have checked the manuscript carefully and reviewed them. Please refer to the revised manuscript.
Point 5: In abstract section, some numerical results should be incorporated.
Response 5: Thank you for your comment.This section was modified according to your comments. Please refer to the revised manuscript.
Point 6: Please add your motivation and contributions of the work in introduction section.
Response 6: We appreciate it very much for this good suggestion, and we have done it according to your ideas. Please refer to the revised manuscript.
Point 7: I’ll suggest adding an organization paragraph at the end of introduction section.
Response 7: We appreciate it very much for this good suggestion, We have added an organization paragraph at the end of introduction section. Please refer to the revised manuscript.
Point 8: The title of the manuscript and some titles of the section are confusing. I think title should be changed. Remember that, you are using NM-PSO for solving the problem. Similarly, Section 5 (Model Construction Based on NM-PSO Algorithm), how model construction can be based on method? Use this section to model your problem mathematically. Write down your objective function and constraints clearly.
Response 8: Thank you for your comment. After our detailed discussion, we have modified the title of Section 5 of the article, and the section name has been changed to(Section 5 Construction of NM-PSO Algorithm).
Point 9: Improve figure 1, 6. Use (a), (b)….. , if there is more than one image in a figure.
Response 9: Thank you for your comment. We have redrawn the figure 1, 6 in the manuscript to ensure clarity of the figure s. In addition, We have revised Figure 1, which upon closer inspection we believe is partially duplicated with Figure 2. Meanwhile, Fig. 6 is the overall flow chart of microseismic source location during coal mining, so (a) and (b) are not used for distinction.
Point 10: Maintain a uniform reference style.
Response 10:Thank you for your comment. We have checked the references carefully and reviewed them, please refer to the references.
Point 11: Please compare your obtained results with few other methods. For example, you can replace PSO by some other algorithms.
Response 11: Thank you for your valuable and thoughtful comments. However, we believe that in the study of microseismic source location in coal mining, NM algorithm is the most widely used. The primary research objective of the manuscript is to further improve the source location accuracy of NM algorithm through PSO algorithm. Therefore, we have not considered the comparison of other algorithms except NM.
Once again, we thank the time you put in reviewing our manuscript. Your comments have been precious.
Round 2
Reviewer 2 Report
good to be accepted now
Reviewer 3 Report
The paper can be accepted in its current form
Reviewer 4 Report
The authors have answered all the queries satisfactorily. I have no more comments excepts some typos which can be taken care at the time of proofread. I think, the paper can be accepted now.